# Classifying Breast Cancer Subtypes Using Multiple Kernel Learning Based on Omics Data

**DOI:** 10.3390/genes10030200

**Published:** 2019-03-07

**Authors:** Mingxin Tao, Tianci Song, Wei Du, Siyu Han, Chunman Zuo, Ying Li, Yan Wang, Zekun Yang

**Affiliations:** 1Key Laboratory of Symbol Computation and Knowledge Engineering of Ministry of Education, College of Computer Science and Technology, Jilin University, Changchun 130012, China; lilytmx18@163.com (M.T.); hansy15@mails.jlu.edu.cn (S.H.); cmzuo13@163.com (C.Z.); liying@jlu.edu.cn (Y.L.); wy6868@jlu.edu.cn (Y.W.); yangzkjlu@foxmail.com (Z.Y.); 2Computational System Biology Laboratory, Department of Biochemistry and Molecular Biology and Institute of Bioinformatics, University of Georgia, Athens, GA 30602, USA; 3Computer Science & Engineering, University of Minnesota, Minneapolis, MN 55455, USA; songtianci1993@hotmail.com

**Keywords:** breast cancer subtypes, MKL, mRNA, methylation data, CNV

## Abstract

It is very significant to explore the intrinsic differences in breast cancer subtypes. These intrinsic differences are closely related to clinical diagnosis and designation of treatment plans. With the accumulation of biological and medicine datasets, there are many different omics data that can be viewed in different aspects. Combining these multiple omics data can improve the accuracy of prediction. Meanwhile; there are also many different databases available for us to download different types of omics data. In this article, we use estrogen receptor (ER), progesterone receptor (PR), human epidermal growth factor receptor 2 (HER2) to define breast cancer subtypes and classify any two breast cancer subtypes using SMO-MKL algorithm. We collected mRNA data, methylation data and copy number variation (CNV) data from TCGA to classify breast cancer subtypes. Multiple Kernel Learning (MKL) is employed to use these omics data distinctly. The result of using three omics data with multiple kernels is better than that of using single omics data with multiple kernels. Furthermore; these significant genes and pathways discovered in the feature selection process are also analyzed. In experiments; the proposed method outperforms other state-of-the-art methods and has abundant biological interpretations.

## 1. Introduction

Breast cancer is the most general cancer diagnosed in women, and it is also the main cause of cancer death behind lung cancer [1]. There are more and more people with breast cancer, and 6.6% of patients are women diagnosed below the age of 40 [2]. Moreover, young women with breast cancer are more likely to have more aggressive subtypes, such as triple-negative or HER2-positive breast cancer, and are more likely to be identified as advanced stages [1]. Meanwhile, breast cancer is also a high heterogeneity disease, and it is comprised of distinct biological subtypes which present a varied spectrum of clinical, pathologic and molecular features with different prognostic and therapeutic implications [3]. The studies on the genotyping of breast cancer are very important for breast cancer treatment decisions and prognosis prediction [4].

Recent studies have been directed at molecular classification of breast cancer. With the development of high-throughput research techniques such as microarray, genotyping can not only reflect the clinical characteristics of distinct types of breast cancer, but also provide technical support for the further study of breast cancer genotyping. The concept of tumor molecular typing was proposed by the National Cancer Institute (NCI) in 1999. Perou et al. in Stanford University firstly reported the molecular classification of breast cancer in 2000, and concluded that there are four primary subtypes including luminal subtype, basal-like subtype, human epidermal growth subtype and normal breast-like subtype [5]. In 2003, Sorlie et al. further divided luminal subtype into luminal A and luminal B [6]. Although there are many other classification methods to define the subtypes of breast cancer, the most widely one is still the method proposed by Perou and Sorlie [5].

The classification of breast cancer subtypes is the basis for the diagnosis, the prognosis, the choice of treatment methods, and the conduct of various studies. In this article, breast cancer was classified into five subtypes based on immunohistochemistry (IHC) markers which include estrogen receptor (ER), progesterone receptor (PR) and human epidermal growth factor receptor 2 (HER2). These subtypes are: luminal A, luminal B, HER2 (+), basal-like/Triple-negative breast cancer (TNBC), and Unclear [7]. The detailed definition of each subtype can be found in Table 1. The Unclear subtypes was the patients which have missing data in ER, PR or HER2, and we collected these patients in the “unclear” group. Luminal A is the most common subtype of breast cancer in clinic, and it is usually an early form of breast cancer. Furthermore, the 5-year local recurrence rate of this subtype is significantly lower than other subtypes [8]. Luminal A subtype patients have a low Ki-67 index, and some studies found cell proliferation activity to be lower in Luminal A than in other breast cancer subtypes [9]. According to reports, the expression of hormone in patients with Luminal B is lower than Luminal A, while the expression of proliferation markers and histologic grade are higher than Luminal A [10]. Probably 25% of breast cancer are classified as HER2-positive, and this subtype has always been related to poor prognosis. Many papers showed that trastuzumab improves the prognosis of HER2-positive breast cancer patients [11]. Basle-like/TNBC is currently the most studied subtype, is easy to deteriorate and metastasize, is highly sensitive to chemotherapy, and more, its prognosis is the worst in five breast cancer subtypes [12]. The incidence of lymph node metastasis is higher than other breast cancer subtypes [13]. The incidence of TNBC is higher among blacks, Amerindians and Asian Indians than in whites.

There are many studies on breast cancer subtypes classification. Brian D. Lehmann et al. used cluster analysis to identify TNBC subtypes based on gene expression [14]. Therese Sørlie et al. classified breast cancer subtypes based on differences in gene expression patterns by hierarchical clustering [15]. There are also many papers on breast cancer subtypes defined by histological information, like Weigelt B. studied a classification of breast cancers of special histological subtypes using immunohistochemistry and genome-wide gene expression data [16]. There are many papers on TNBC subtypes, like Liu YR developed a TNBC classification system using mRNAs and IncRNAs [17]. However, studies on breast cancer that focus on one omics dimension have only provided limited information, and deriving multiple types of omics data from the same patient is increasing common. Meanwhile, more omics databases of high quality have emerged, and they provide us with more possibilities to collect distinct omics data.

Support Vector Machine (SVM) is a supervised learning model, and it has been generally used in classification problems. It effectively addresses non-linearly separable problems by introducing the kernel function to map the features of data to a higher dimension, where data are linearly separable. The classical kernel functions include linear kernel, polynomial kernel and gaussian kernel. Subsequently, we can use Multiple Kernel Learning (MKL) to improve the classification accuracy of SVM [18]. MKL can be used in two approaches: (a) Various kernels correspond to various notions, therefore using multiple kernels can reduce bias. (b) Different kernels may use different representations of input from different sources [18,19]. In our article, we used SMO-MKL to classify breast cancer subtypes, and this method can produce a better result than using MKL [20].

SMO-MKL is an improved supervised method based on linear MKL framework fusing heterogeneous omics data of breast cancer from the Cancer Genome Atlas (TCGA) [20]. The workflow of this method contains two parts, is showed in Figure 1 as below. We mainly used MKL fusing heterogeneous omics data to predict subtypes of breast cancer. Firstly, we collected distinct omics data of breast cancer from TCGA, including mRNA data, DNA methylation data and Copy Number Variation (CNV) data, and only selected the patients with these omics data and subtypes information simultaneously. Secondly, we processed these data by purging and normalizing. To solve the redundant problem in omics data, we used a gene feature selection method based on different omics data, respectively. Finally, we generated kernels using omics data as an input of the subsequent model, and then trained the SMO-MKL model to obtain the classification result of the predictor.

## 2. Methods

### 2.1. Data Sources

In this article, we collected breast cancer data from The Cancer Genome Atlas (TCGA) (http://cancergenome.nih.gov/), which is a public database containing thousand samples of cancer patients, which profiles the various omics data of about 10,000 patient samples across over 30 cancer subtypes [21]. Among the various types of data, we selected mRNA, DNA methylation and Copy number variation data to validate the proposed method, because these three types of data have been measured on most patient samples simultaneously. We also used the information of ER, PR and Her2 from clinical data to determine the corresponding subtypes of cancer patient samples [7]. The breast cancer subtypes were classified into five groups using ER, PR and Her2. Our dataset contained 606 distinct patient samples of breast cancer, which was divided into five subtypes: 277 luminal A, 40 luminal B, 70 Triple Negative Breast Cancer (TNBC), 11 HER2 (+), and 208 unclear. The data were shown in Table 2. In this study, we acquired the RNA sequencing level 3 data as mRNA data, DNA Methylation 450k level 3 data as DNA methylation data, and the Affymetrix SNP 6.0 array data with GRCH 38 (hg38) genome data as CNV data.

Furthermore, we normalized these data before performing feature selection. For mRNA data, we used the original mRNA data from TCGA, and removed the genes which have missing values over 200 samples. As for methylation data, we only considered regulatory relationships between the gene transcription and the hyper-methylation or hypo-methylation of relevant promoter. Moreover, we matched probes in methylation data to gene using these relationships. For CNV data, we also firstly converted the ID of probes to gene symbols by CNV annotation in PennCNV [22], and then merged corresponding values based on mapping between probes and genes. Eventually, each omics data was represented as a two-dimensional matrix with each row representing one gene symbol and each column representing one sample with its own corresponding subtype.

### 2.2. Feature Selection

Because some genes may have few or even no effect on classifying the subtypes of breast cancer, we selected some significant genes by applying feature selection techniques. In general, the feature selection methods can be divided into three categories: filter methods, wrapper methods, and embedded methods [23]. In our article, we used an advanced filter method to select the informative genes in these three omics data respectively. The reason why we chose the filter method is that we can access all informative genes instead of keeping one gene but omitting other genes with consistent patterns. Firstly, we used Wilcoxon rank-sum test on each omics data to obtain the p-values of genes separately. Then we chose the Benjamini-Hochberg False Discovery Rate (BH-FDR) to adjust these *p*-values. In this article, we selected the gene with p-values less than 0.05 as the significant gene.

### 2.3. The Process of Classification

#### 2.3.1. Multi-Omics Data Fusing

For breast cancer subtypes prediction, we generated kernels of MKL on mRNA data, methylation data and CNV data separately. Because the different scales of different omics data are distinct, so we normalized these kernels using the formula as below:(1)$$K_{norm}\left(X_{i},X_{j}\right)=\frac{K\left(X_{i},X_{j}\right)}{\sqrt{K\left(X_{i},X_{i}\right)K\left(X_{j},X_{j}\right)}}$$
where K indicates kernel function, $X_{i}$ represents the i-th data point and $X_{j}$ represents the *j*-*th* data point.

In this article, we used three types of kernel functions, concluding Linear kernel $K\left(x_{i},x\right)=\left(x_{i},x\right)$, Gaussian kernel $k\left(x_{i},x\right)=exp\left(-\frac{x-y^{2}}{2\sigma^{2}}\right)$ and Polynomial kernel $K\left(x_{i},x\right)={\left[\left(x_{i},x\right)+1\right]}^{2}$. In the practical application, different kernels can be combined together. Linear kernel is enough if the data can be linearly separated, and radial basis function kernel and Polynomial kernel are used to cope with the non-linear separable problem.

#### 2.3.2. SMO-MKL Classification

Support Vector Machine is a quadratic programming (QP) problem, and it needs to consume lots of time and memory to solve the QP problem. Sequential minimization optimization (SMO) is an efficient and easily applicable optimization method. SMO can resolve large QP problem into a series of smallest QP problems. Each resolved QP problems only have two Lagrange multipliers, so this can reduce the complexity of the large QP problem [20].

Multiple Kernel Learning algorithms are extensively used in multi-omics data fusion to improve the performance and interpretation [19]. In this article, we used an improved MKL regularized with $l_{p}$ norm, using the Sequential Minimal Optimization (SMO), which is easy to implement and efficiently scales to large problems [20]. MKL using ℓ-*norm* regularization can lead to a sparse solution, and MKL which is the dual of standard p-norm is differentiable, and it can lead to a dense solution on kernel weight. SMO-MKL used the dual of standard p-norm MKL, and it utilized SMO style co-ordinate ascent to optimize. SMO-MKL can get a better result than original MKL using l-norm regularization [20]. The kernels which encode different omics data are determined to be an optimal linear combination of given base kernels with non-negative weights.

## 3. Results

In this part, we firstly showed the accuracy and the area under the curve (AUC) of classification on any two breast cancer subtypes. Then, we achieved multi-classification by integrating multiple binary classifiers and compared with random forest with our model base on same datasets. Finally, we analyzed some significant genes and pathways.

### 3.1. Comparison Binary Classification by Different Omics

We combined different types of omics data to classify any two subtypes of breast cancer including: (1) luminal A versus luminal B, (2) luminal A versus TNBC, (3) luminal A versus HER2 (+), (4) luminal A versus Unclear, (5) luminal B versus TNBC, (6) luminal B versus HER2 (+), (7) luminal B versus Unclear, (8) TNBC versus HER2 (+), (9) TNBC versus Unclear, (10) HER2 (+) versus Unclear. We used mean accuracies of 10-fold cross-validation and AUCs as measurements of MKL model. Additionally, we calculated accuracies and AUCs with combining different kernels and omics data. These classification performances were shown in Table 3 and Table 4. Table 3 showed that the best accuracies were achieved by our model in most cases. Table 4 showed that the best AUCs are achieved by the model we used in most cases. Moreover, we plotted the ROC of these results in Appendix A.

We can get the result that the optimal performance is Luminal A & HER2 (+) in accuracy and AUC. The MKL accuracy of HER2 (+) with other subtypes were all better than single omics data. The accuracy of the Luminal A with HER2 (+) and Unclear are better than Luminal B with these two breast cancer subtypes. Meanwhile, several studies are shown that the expression of ER-related genes in patients with Luminal A are higher than patients with Luminal B [24,25]. We also used a boxplot to show the gene expression of ESR1 in Luminal A and Luminal B, and it was shown in Figure 2. As such, we speculated that the reason was related to the expression of ER-related genes. The worst performance in these classifications is HER2 (+) & TNBC. We conjectured the main reason for this is that the patient numbers in these two subtypes are fewer than for other subtypes.

### 3.2. Comparison Multi-Classification by Different Omics

We also used this algorithm to predict breast cancer subtypes. Because the method we used is a binary classification algorithm, we achieved multi-classification by integrating multiple binary classifiers. There are two strategies for turning to multinomial classifiers: One vs rest (OvR) and One vs one (OvO). We used OvO to turn this method to multi-classification since its, low sensitivity to the imbalanced data. The theory of OvO is to train k/(k−1) binary classifiers, and predict through voting. In our experiment, the results were achieved by 10-fold cross-validation. The accuracy of multi-classification was shown in Figure 3, and we also calculated precision and recall of multi-classification, and these two tables in Appendix A. We demonstrated that MKL was superior to those models with single omics data in the breast cancer subtypes prediction.

### 3.3. Comparison Subtypes of Triple-Negative Breast Cancer Multi-Classification

Lehmann et al. used K-means clustering to classify TNBC patients into six clusters using gene expression data, and then defined these six clusters as six TNBC subtypes [14]. These TNBC subtypes are: (1) two basal-like (BL1 and BL2) subtypes; (2) immunomodulatory (IM) subtype; (3) mesenchymal (M) subtype; (4) mesenchymal stem-like (MSL) subtype and luminal androgen receptor (LAR) subtype. Now authors provided a web-based subtyping tool (http://cbc.mc.vanderbilt.edu/tnbc/) [26], and we can use this web-based algorithm to obtain the subtype of our samples. Because our datasets only have 70 TNBC subtypes, so some TNBC subtypes numbers obtained from this tool were less than 10. Considering classification accuracy, we only chose some TNBC subtypes (BL1, IM and M) which patient numbers more than 10. We used the same multi-classification method with Section 3.2 to multi-classify TNBC subtypes, and the result was shown in Figure 4.

### 3.4. Comparison with Other Methods

We compared SMO-MKL with Random Forest and Neural Network on the same datasets. The reasons we chose Random Forest as compared method was that we can use random forest to utilize different omics data indirectly through its tree structure, and RF often used in multiple biological data integration [27]. The Random Forest was used random Forest R-package. The reason we chose Neural Network was that Neural Network has a good performance in classification. The Neural Network was used with the RSNNS R-package.

To better compare the performance of Random Forest and Neural Network with SMO-MKL, we applied the same settings in our model to random forest and Neural Network, and the result was achieved by 10-fold cross-validation of these approaches. The accuracy and AUC were shown in Figure 5 and Figure 6. In Figure 5 and Figure 6, the almost classification results of SMO-MKL were better than Random Forest. The reason that some accuracies of Neural Network were better than SMO-MKL is that some binary classifications have an unbalance problem, and Neural Network completely classified one class, so the AUC of these binary classifications was low.

### 3.5. Analysis of Selected Genes

In this part, Firstly, we displayed the heatmap depicted by top 10 differential genes, and analyzed these significant genes further. Secondly, we analyzed pathways enriched by genes. 

#### 3.5.1. Heatmap of Selected Genes

The top 10 genes are ranked by p-values in decreasing order in each omics data. Heatmap is utilized to show all top 10 differential genes in any two classifications between Luminal A, Luminal B, TNBC and HER2 (+). These heatmap results were shown in Figure 7, Figure 8 and Figure 9. From these three heatmap, the breast cancer subtypes were classified more clearly in mRNA data. When we selected these genes, we found some genes were significant in many classifications at the same time, meanwhile, some important genes were also appeared in several omics data, like ERBB2.

Additionally, some significant genes were also selected to make further interpretation. In the classification of Luminal A versus Luminal B, the MED1 gene was related to HER2 (+) status of cancer in tissue microarray analysis [28]. Furthermore, MED1 can inhibit tamoxifen in clinical therapy. In the investigation of tamoxifen in treatment, we found that patients with Luminal A will have more long-term benefits than Luminal B [29]. The GATA3 expression was different in Luminal A versus TNBC and Luminal B versus TNBC, and its expression level was the higher in ER-positive breast cancer than other breast cancer subtypes. The expression of GATA3 was positively correlated with ER-positive breast cancer subtypes [30,31]. Gene STARD3 was related to HER2 amplification in Luminal A versus HER2 (+) and HER2 (+) versus TNBC, and it linked to growth and survival of HER2-amplified cancer cells. At the same time, STARD3 gene has no function in other cancer cells [32]. Stefka et al. used proteins to find GRB7 gene had an intimate connection with HER2-enriched breast cancer and was amplified in Her2 positive breast cancer. Moreover, the study also indicated that the GRB7 gene is clearly different between her2-positive breast cancer and other breast cancer subtypes [33]. It was found in classifying Luminal A versus Luminal B, Luminal A versus HER2 (+) and HER2 (+) versus TNBC. The XBP1 gene had an important role in the tumorigenicity in breast cancer subtypes and had higher expression level and activity in TNBC than luminal. It was found in classifying Luminal B versus TNBC [34]. The ESR1 gene was the most significantly different gene in classifying Luminal A versus TNBC from mRNA data. The ESR1 gene encodes the estrogen receptor, and some studies try to use the ESR1 gene as a marker gene in breast cancer therapy. Because ER is positive in lots of breast cancer subtypes, they wanted to use this marker to conduct hormonal treatment [35,36,37].

#### 3.5.2. Enrichment of Selected Genes

To gain more insights into the differences in breast cancer subtypes, we separately performed enrichment analysis on the differential genes screened out from the omics data in any two subtypes between Luminal A, Luminal B, TNBC and HER2 (+). The pathways we mainly used were from KEGG and REACTOME. Moreover, we used the P-value obtained from the hypergeometric distribution to sort the pathways and selected top 30 to analysis categories of these pathways (see Appendix A). Although genes were selected by different classifiers, the pathways were concentrated in several categories. Most significant pathways were related to four types: cell cycle, signaling, metabolism, and immune. Cell cycle is the series of events that take place in a cell leading to its division, and the cell cycle pathway can help to display cell growth of the lowest level, the signaling pathway suggested the regulation on cell division. Thus, cell cycle and signaling pathways suggested difference cell growth between distinct breast cancer subtypes. The metabolic pathways are a series of chain reactions that occur within the cell. Metabolic pathways and immune pathways can reflect the differences at the organizational level. The pathways related to cell cycle like REACTOME_CELL_CYCLE and KEGG_CELL_CYCLE; pathways related to signaling like REACTOME_SIGNALLING_BY_NGF, and REACTOME_SIGNALING_BY_THE_B_CELL_RECEPTOR_BCR; the pathways related to metabolic like REACTOME_METABOLISM_OF_LIPIDS_AND_LIPOPROTEINS and REACTOME_METABOLISM_OF_AMINO_ACIDS_AND_DERIVATIVES. The pathways connected immune like REACTOME_IMMUNE_SYSTEM and REACTOME_ADAPTIVE_IMMUNE_SYSTEM. Although these four types of pathways were repeated in different classifications, the numbers were distinct, like in Luminal A versus Luminal B, the enriched pathways were mainly cell cycle pathways, and in Luminal A versus TNBC, the enriched pathways were mainly signaling pathways.

## 4. Discussion

Breast cancer has many distinct subtypes, and these subtypes have different biological, clinical and molecular characteristics [38,39]. We also learned from articles and clinicians that the treatments and drugs options for distinct subtypes of breast cancer patients are different, so accurately identifying breast cancer subtypes is very important. Moreover, identifying the subtypes can help doctors to develop the optimal treatment plan for patients [4].

In our article, the mRNA, methylation and CNV data from TCGA can efficiently classify the breast cancer subtypes using ER, PR, HER2 defined. We also used Random Forest and Neural Network to compare the result with SMO-MKL, and we found that the performance of SMO-MKL was superior to Random Forest and Neural Network. We further applied the SMO-MKL model to predict breast cancer subtypes, and the result also showed that integrated multi-omics data can better separate breast cancer subtypes. In conclusion, the result of any two breast cancer subtypes indicated the multiple omics data can improve the accuracy and AUC in breast cancer subtypes classification using SMO-MKL.

The parameters in polynomial kernel and gaussian kernel also could affect the performance of breast cancer subtypes prediction, which meant that we could improve the accuracy through rationally tuning parameters. As such, we used grid search to select the optimal parameters in polynomial kernel and Gaussian kernel individually in any two breast cancer subtypes.

We searched many papers related to the genes which were have lower p-value in feature selection, and we found many genes related to breast cancer subtypes. and we also observed some selected genes shared by two or more classifications, like GATA3 and GRB7. GATA3 is overexpressed in luminal epithelial cells, and low expression of GATA3 is connected negative ER and PR, so it showed significantly different expression in Luminal A versus TNBC and Luminal B versus TNBC. GRB7 is associated with the status of Her2. We can clearly distinguish the status of Her2 in tumors under the protein expression of GRB7 [33]. 

Moreover, we can further boost our model’s performance in classifying breast cancer subtypes which are based on ER, PR and Her2. We only simply classified breast cancer subtypes utilizing ER, PR and Her2, but they also have some other factors in the clinical diagnosis, like Ki-67 [40]. Therefore, we can collaborate with clinicians to improve the accuracy of the sample labels by adding other clinical indicators. We also collected histological information of breast cancer patients in our datasets from TCGA clinical PDF files. But since most patients in our datasets had Invasive ductal carcinoma, this situation caused a serious unbalance of patient numbers in different histological types. Therefore, we can collect additional data, and do this analyzation in our future research. At present, since the number of samples in our experiment are not very high, especially for HER2 (+), which is one aspect that limited our analysis and affects our accuracy, we can try to collect more samples from different sources for further analysis. Since patients with different breast cancer subtypes have different responses to distinct chemotherapeutic drugs and other treatment methods, we can collect more subtype-specific genes from papers, and gain more insight into the differences between subtypes [39,41]. Also, now there are also many studies on searching maker genes, which almost all used mRNA, miRNA, IncRNAs, SNP and proteomic data, and some used pathway [36,42,43,44]. We can try to combine these omics data and pathway in our analysis. Therefore, we think we can conduct further study from this perspective.

## 5. Conclusions

The purpose of this study was to classify breast cancer subtypes by using SMO-MKL models and multiple omics datasets from TCGA, and then analyze differences between distinct breast cancer subtypes in many aspects. The classification results showed that integrating multiple omics datasets can improve the accuracy than using single omics data through SMO-MKL models. Furthermore, we achieved multi-classification on breast cancer subtypes and TNBC subtypes by using SMO-MKL models, and the classification results also showed that the accuracy of using multiple omics data were higher than single omics data. Meanwhile, we tried to give some biological explanation for the differences between breast cancer subtypes by analyzing the significant genes and pathways, which also provide guidance for exploring the biological models on breast cancer subtypes.

## Figures and Tables

**Figure 1 genes-10-00200-f001:**
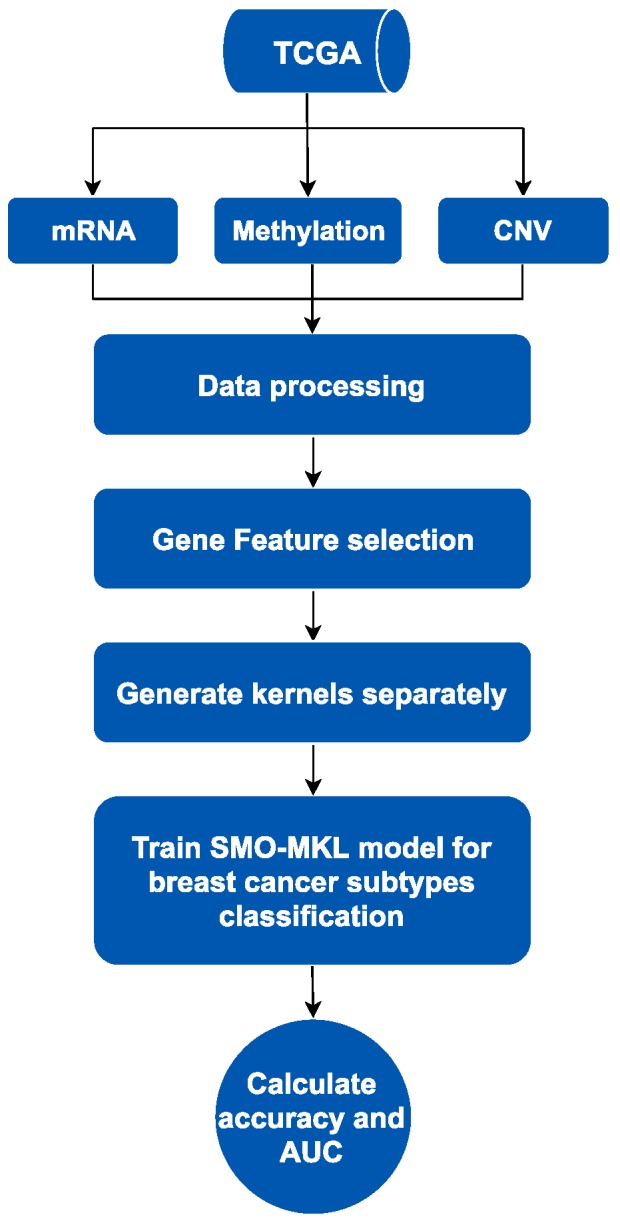
The process of breast cancer subtypes prediction. TCGA: The Cancer Genome Atlas; CNV: copy number variation.

**Figure 2 genes-10-00200-f002:**
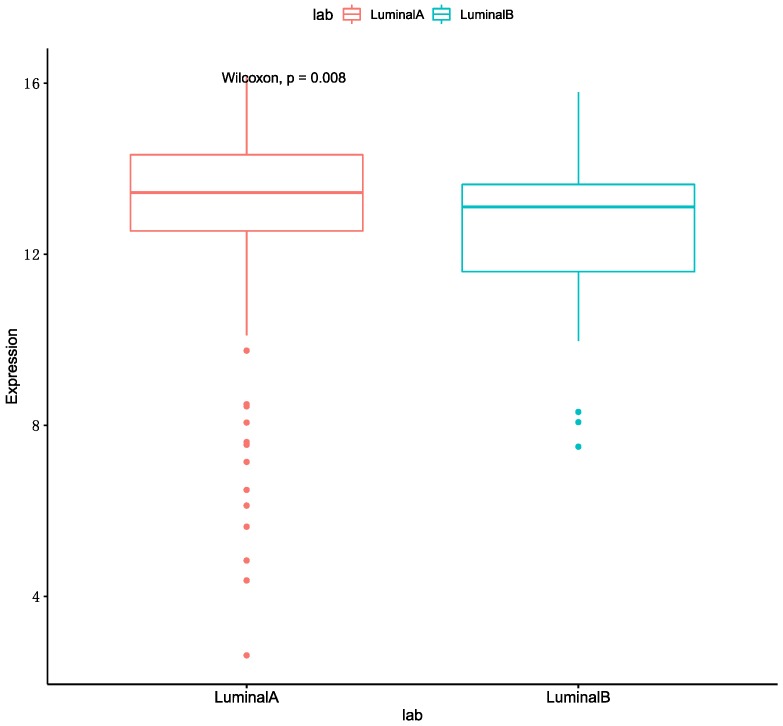
The gene expression of ESR1.

**Figure 3 genes-10-00200-f003:**
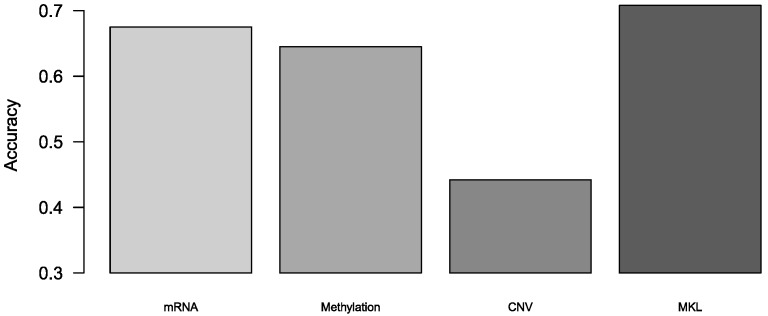
The accuracy of multi-classification in breast cancer subtypes.

**Figure 4 genes-10-00200-f004:**
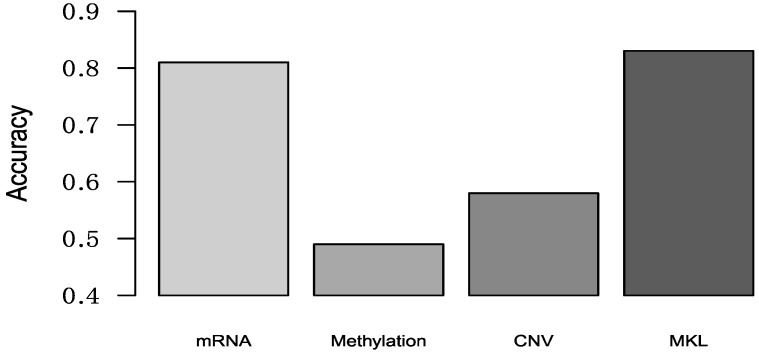
The accuracy of multi-classification in TNBC subtypes.

**Figure 5 genes-10-00200-f005:**
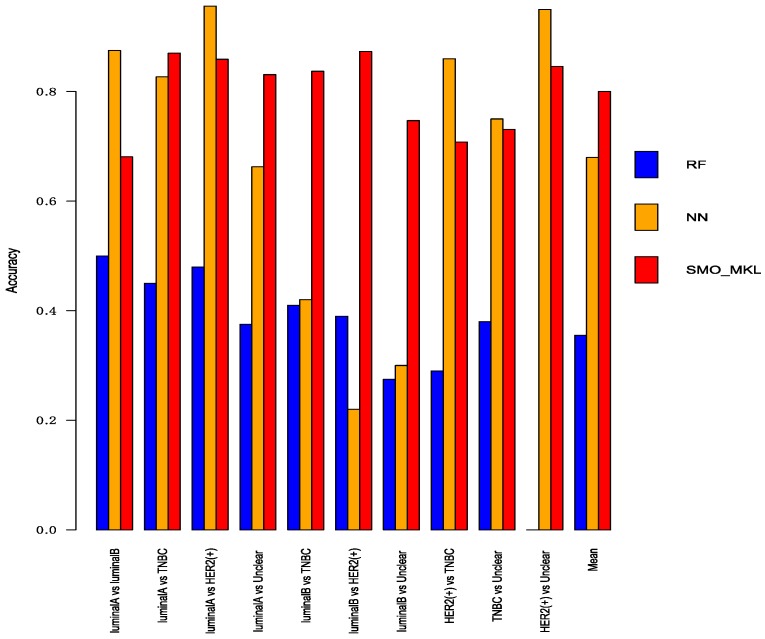
The Accuracy of Random Forest, Neural Network, SMO-MKL on any two breast cancer subtypes.

**Figure 6 genes-10-00200-f006:**
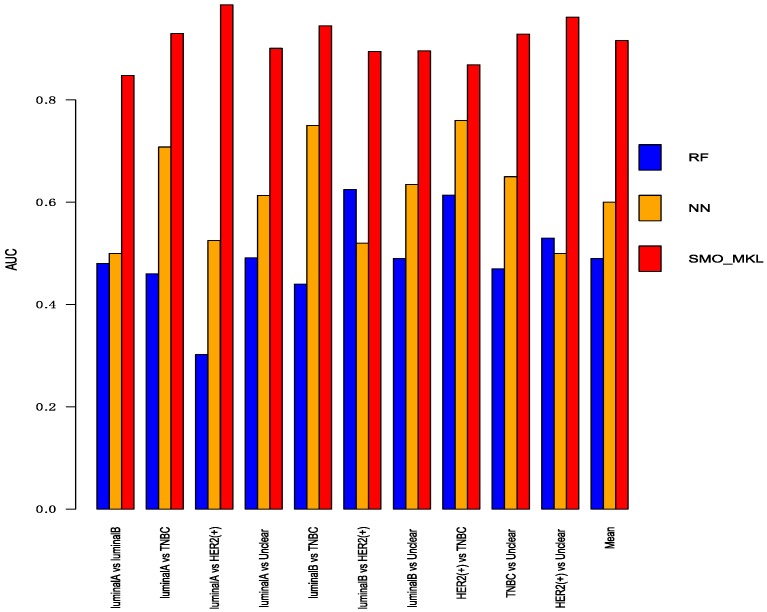
The AUC of Random Forest, Neural Network, SMO-MKL on any two breast cancer subtypes.

**Figure 7 genes-10-00200-f007:**
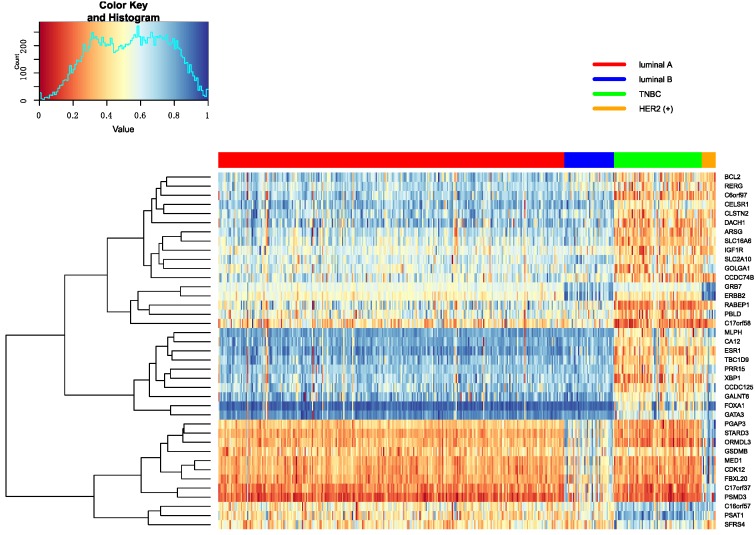
The heatmap of breast cancer subtypes in mRNA data.

**Figure 8 genes-10-00200-f008:**
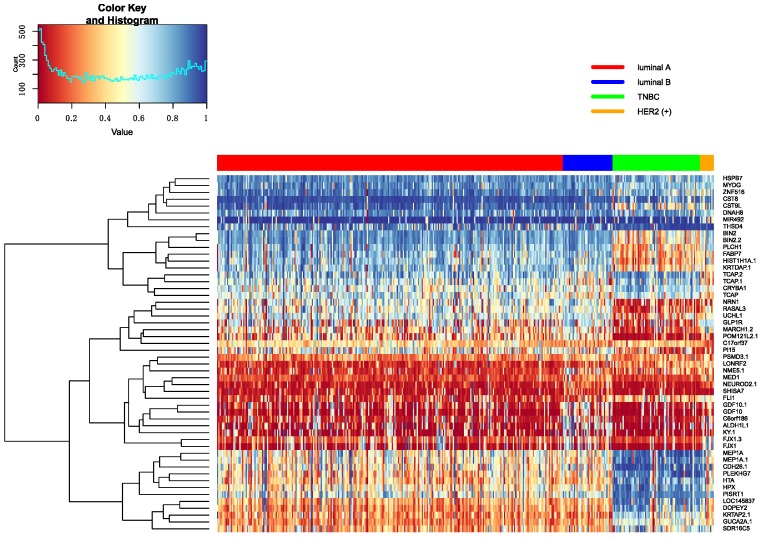
The heatmap of breast cancer subtypes in Methylation data.

**Figure 9 genes-10-00200-f009:**
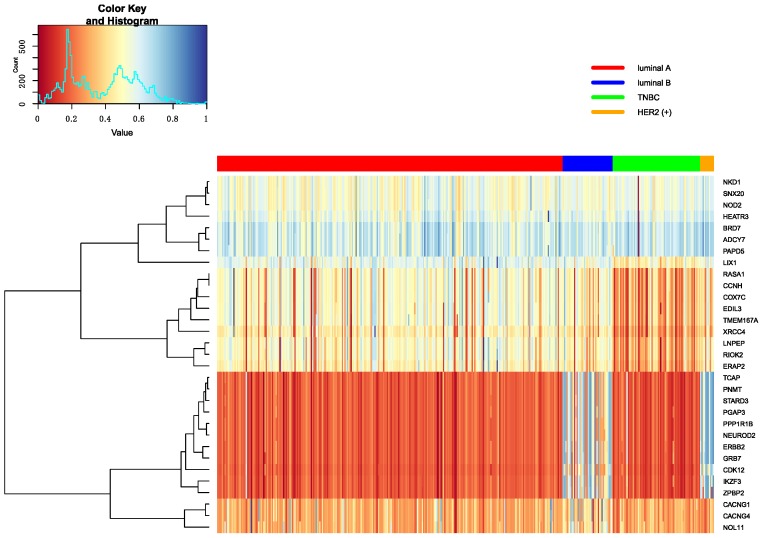
The heatmap of breast cancer subtypes in CNV data.

**Table 1 genes-10-00200-t001:** The definition of distinct breast cancer subtypes. TNBC: Triple-negative breast cancer; HER2: human epidermal growth factor receptor 2.

Breast Cancer Subtypes	Definition
Luminal A	ER/PR+, Her2−
Luminal B	ER/PR+, Her2+
TNBC	ER/PR−, Her2−
HER2 (+)	ER/PR−, Her2+
Unclear	Other samples

**Table 2 genes-10-00200-t002:** The numbers of distinct breast cancer subtypes.

Breast Subtypes	Cancer Patients
Luminal A	277
Luminal B	40
TNBC	70
HER2 (+)	11
Unclear	208

**Table 3 genes-10-00200-t003:** The accuracies of any two breast cancer subtypes with three kernels. Bold numbers are the best performance of this binary classification.

Breast Cancer Subtypes	mRNA	Methylation	CNV	MKL
Luminal A vs. luminal B	0.436	0.436	0.490	**0.681**
Luminal A vs. HER2 (+)	0.739	0.566	0.739	**0.870**
Luminal A vs. TNBC	**0.868**	0.867	0.604	0.859
Luminal A vs. Unclear	0.760	**0.849**	0.473	0.831
Luminal B vs. HER2 (+)	0.732	0.776	0.485	**0.837**
Luminal B vs. TNBC	0.871	**0.883**	0.855	0.873
Luminal B vs. Unclear	0.696	0.748	**0.770**	0.747
HER2 (+) vs. TNBC	0.5	0.5	0.5	**0.708**
HER2 (+) vs. Unclear	0.495	0.498	0.5	**0.731**
TNBC vs. Unclear	0.806	0.836	0.717	**0.846**
Mean	0.690	0.696	0.613	**0.798**

**Table 4 genes-10-00200-t004:** The AUC of any two breast cancer subtypes with three kernels. Bold numbers are the best performance of this binary classification.

Breast Cancer Subtypes	mRNA	Methylation	CNV	MKL
Luminal A vs. luminal B	0.835	0.632	0.810	**0.848**
Luminal A vs. HER2 (+)	0.973	0.903	0.979	**0.986**
Luminal A vs. TNBC	**0.934**	0.926	0.909	0.930
Luminal A vs. Unclear	0.824	0.878	0.589	**0.901**
Luminal B vs. HER2 (+)	0.843	0.824	0.725	**0.895**
Luminal B vs. TNBC	**0.947**	0.932	0.941	0.945
Luminal B vs. Unclear	0.875	0.808	0.835	**0.896**
HER2 (+) vs. TNBC	0.867	0.778	0.741	**0.869**
HER2 (+) vs. Unclear	0.925	0.873	0.859	**0.962**
TNBC vs. Unclear	0.902	0.918	0.834	**0.929**
Mean	0.893	0.847	0.822	**0.916**

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
