# Peer review of "Classifying Breast Cancer Subtypes Using Multiple Kernel Learning Based on Omics Data"

_genes, 2019, doi:10.3390/genes10030200_

Round 1

Reviewer 1 Report

 The authors used a SMO-MKL models and multiple omics data to classify breast cancer subtypes. And the accuracy and AUC they got was shown that integrating three omics data were all better than using single omics data by using the SMO-MKL models, then they analyzed the significant genes and pathways based on feature selection. However, there are some issues should attend.
1)      The authors should describe the SMO-MKL method and the difference between distinct breast cancer subtypes in more details.
2)      The authors only compared SMO-MKL model to Random Forest on the same breast cancer database. They should better compare to some other popular feature selection algorithms in the same time.
3)      The authors should show more significant differential genes between distinct breast cancer subtypes based on feature selection, and analyze the enrichment pathway of them by some table lists.
4)      Figure 2 is too simple. The authors should redraw it and add more explanation.
5)      Please recheck the manuscript and correct the grammar mistakes. Here are some typos:
(1)    In line 5 on page 5, “… Probably 25% of breast cancer are classified as HER2-positive, and this subtype has always been related with poor prognosis”, the “with” can corrected as “to”.
(2)    In line 14 on page 20, “…display cell growth from the lowest level…”, the “from” can be corrected as “of”.
(3)    In line 14 on page 23, “… the differences among subtypes…”, the “among” can be corrected as “between”.

Author Response

Response to the Editor and Reviewers’ Comments Title: Classifying Breast Cancer Subtypes using Multiple Kernel Learning based on Omics Data Submitted to Genes The authors would like to thank the editor and reviewers for the valuable comments and constructive suggestions to this manuscript. Please see our responses to the editor and reviewers and modifications being made following the editor and reviewers’ comments. Response to Reviewer #1: Comment 1: The authors should describe the SMO-MKL method and the difference between distinct breast cancer subtypes in more details Response: We have described the SMO-MKL method in the section 2.3.2 SMO-MKL Classification. And we have added more difference introduction of distinct breast cancer subtypes in the section Introduction. Comment 2: The authors only compared SMO-MKL model to Random Forest on the same breast cancer database. They should better compare to some other popular feature selection algorithms in the same time. Response: We now added neural network compared with SMO-MKL model, we have updated that in the section 3.3 Comparison with other methods. Comment 3: The authors should show more significant differential genes between distinct breast cancer subtypes based on feature selection, and analyze the enrichment pathway of them by some table lists. Response: We shown the enrichment pathway in supplement materials Comment 4: Figure 2 is too simple. The authors should redraw it and add more explanation. Response: We have re-drew the Figure 2 using the R package ggplurs. Comment 5: Please recheck the manuscript and correct the grammar mistakes. Here are some types: 1) In line 5 on page 5, “… Probably 25% of breast cancer are classified as HER2-postive, and this subtypes has always been related with poor prognosis”, the “with” can corrected as “to”. 2) In line 14 on page 20, “… display cell growth from the lowest level…”, the “from” can be corrected as “of”. 3) In line 14 on page 23, “… the differences among subtypes…”, the “among” can be corrected as “between”. Response: We have rechecked the manuscript and corrected these mistakes in the manuscript.

Reviewer 2 Report

Tao and coworkers present the manuscript entitled “Classifying Breast Cancer Subtypes using Multiple 2 Kernel Learning based on Omics Data”. As a breast cancer researcher I found this study very useful. It was well performed and described and deserves publication. However some important concerns should be fully addressed before potential publication in Genes.

Main concerns

Introduction
Several important citations are missing. Diverse approaches in breast cancer classification and new subtypes have been reported, thus they should summarized in Introduction section, e.g.: Liu YR. Comprehensive transcriptome analysis identifies novel molecular subtypes and subtype-specific RNAs of triple-negative breast cancer. Breast Cancer Res. 2016 Mar 15;18(1):33.  Abramson VG. Subtyping of triple-negative breast cancer: implications for therapy. Cancer. 2015 Jan 1;121(1):8-16. Vuong D. Molecular classification of breast cancer. Virchows Arch. 2014  Jul;465(1):1-14. Weigelt B. Refinement of breast cancer classification by molecular characterization of histological special types. J Pathol. 2008 Oct;216(2):141-50. Dawson SJ. A new genome-driven integrated classification of breast cancer and its implications. EMBO J. 2013 Mar 6;32(5):617-28. Weigelt B. Refinement of breast cancer classification by molecular characterization of histological special types. J Pathol. 2008 Oct;216(2):141-50., etc.

Page 3: line 81 “In our article, we used SMO-MKL...”. Please define SMO.

Page 4, line 104. “Our dataset 104 contained 606 distinct patient samples of breast cancer, which was divided into five subtypes: 277 105 luminal A, 40 luminal B, 70 Triple Negative Breast Cancer (TNBC), 11 HER2 (+), and 208 unclear. ”. Please better define the characteristics of the “unclear” group.

Results
The main criticism to the study is about the design. Although current molecular classification of breast cancer is based on the outstanding papers from Perou and Sorlie, many other new subtypes of breast cancer have been described recently. Why authors did not considered the new triple negative breast cancer subtypes in their analyses ?, for instance: i) the immunomodulatory subtype (IM), ii) the luminal androgen receptor subtype (LAR), iii) the mesenchymal-like subtype (MES) and iv) the basal-like and immune suppressed (BLIS) subtypes ?. They are recognized subtypes breast cancer and should be included in the analysis showed in this manuscript even when they were not classified in TGCA. I encourage authors to perform an analysis to include the new subtypes of breast cancer in this study.

Current therapy decision-making is governed by the molecular classification of breast cancer (luminal, basal-like, HER2+). The molecular classification is derived from mainly invasive ductal carcinoma not otherwise specified (IDC NOS), whereas about 25% are defined as histological 'special types'. Authors should refine the breast cancer classification systems by showing the histological types, as well as the special types [invasive lobular carcinoma (ILC), tubular, mucinous A, mucinous B, neuroendocrine, apocrine, IDC with osteoclastic giant cells, micropapillary, adenoid cystic, metaplastic, and medullary carcinoma] derived from immunohistochemistry and genome-wide gene expression profiling.

Multi-classification on breast cancer subtypes by using SMO-MKL models in this study, should be able to classification also the new subtypes of breast cancer based on omics data.

Finally, limitations of this study should be clearly stated. For instance, TNBC classification system integrating the expression profiles of mRNAs, but not lncRNAs, mutations or proteomic data.

Author Response

Response to the Editor and Reviewers’ Comments Title: Classifying Breast Cancer Subtypes using Multiple Kernel Learning based on Omics Data Submitted to Genes The authors would like to thank the editor and reviewers for the valuable comments and constructive suggestions to this manuscript. Please see our responses to the editor and reviewers and modifications being made following the editor and reviewers’ comments. Response to Reviewer #2: Comment 1: Introduction Several important citations are missing. Diverse approaches in breast cancer classification and new subtypes have been reported, thus they should summarized in Introduction section, e.g.: Liu YR. Comprehensive transcriptome analysis identifies novel molecular subtypes and subtype-specific RNSs of triple-negative breast cancer. Breast Cancer Res. 2016 Mar 15;18(1):33. Abramson VG. Subtyping of triple-negative breast cancer: implications for therapy. Cancer. 2015 Jan 1;121(1): 8-16. Vuong D. Molecular classification of breast cancer. Virchows Arch. 2014 Jul; 465(1):1-14. Weigelt B. Refinement of breast cancer classification by molecular characterization of histological special types. J Pathol. 2008 Oct;216(2):141-50. Dawson SJ. A new genome-driven integrated classification of breast cancer and its implications. EMBO J. 2013 Mar 6;32(5):617-28. Weigelt B. Refinement of breast cancer classification by molecular characterization of histological special types. J Pathol. 2008 Oct;216(2):141-50., ect. Response: We have read carefully and thoroughly these papers and have modified this section following your suggestions. Comment 2: Introduction Page 3: line 81 “In our article, we used SMO-MKL…”. Please define SMO. Response: We have modified this section following your suggestions. Comment 3: Introduction Page 4, line 104. “Our dataset 104 contained 606 distinct patient samples of breast cancer, which was divided into five subtypes: 277 105 luminal A, 40 luminal B, 70 Triple Negative Breast cancer (TNBC), 11 HER2 (+), and 208 unclear.”. Please better define the characteristics of the “unclear” group. Response: The “unclear” group in our manuscript is the patients which have missing data in ER, PR or HER2, and we collected these patients in the “unclear” group. We also added detailed description about “unclear” group in the manuscript. Comment 4: Results The main criticism to the study is about the design. Although current molecular classification of breast cancer is based on the outstanding papers from Perou and Sorlie, many other new subtypes of breast cancer have been described recently. Why authors did not considered the new triple negative breast cancer subtypes in their analysis? for instance: i) the immunomodulatory subtype (IM), ii) the luminal androgen receptor subtype (LAR), iii) the mesenchymal-like subtype (MES) and iv) the basal-like and immune suppressed (BLIS) subtypes? They are recognized subtypes breast cancer and should be included in the analysis showed in this manuscript even when they were not classified in TCGA. I encourage authors to perform an analysis to include the new subtypes of breast cancer in this study. Response: We have searched many papers related the new triple negative breast cancer subtypes and downloaded the clinical data in TCGA to find ways defined new triple negative breast cancer, and found a web-based subtyping tool TNBCtype for candidate TNBC subtypes using our gene expression data, and then we used the classification result from this tool to do multi-classification of TNBC subtypes. Because we only have 70 TNBC samples in our datasets, so patients numbers concluded in some TNBC subtypes less than 10 patients, and these will have bad influences on results based on this, furthermore, we can not use 10-fold cross-validation. So we only chose TNBC subtypes which patients more than 10 to do multi-classification. The method of multi-classification is the same with multi-classification in our manuscript. And we have added these experimental results to Result section. Comment 5: Results Current therapy decision-making is governed by the molecular classification of breast cancer (luminal, basal-like, HER2+). The molecular classification is derived from mainly invasive ductal carcinoma not otherwise specified (IDC NOS), whereas about 25% are defined as histological ‘special types’. Authors should refine the breast cancer classification systems by showing the histological types, as well as the special types [invasive lobular carcinoma (ILC), tubular, mucinous A, mucinous B, neuroendocrine, apocrine, IDC with osteoclastic giant cells, micropapillary, adenoid cystic. Metaplastic, and medullary carcinoma] derived from immunohistochemistry and genome-wide gene expression profiling. Multi-classification on breast cancer subtypes by using SMO-MKL models in this study, should be able to classification also the new subtypes of breast cancer based on omics data. Response: We have searched many papers related to the new subtypes of breast cancer, and we found PDF of clinical data from TCGA. It has the histological types of patients in clinical data PDF files. So we tried to use these information as labels to multi-classify the breast cancer. But when we collected histological information from these PDF files, most patients were Invasive ductal carcinoma, and patient number in most histological types were less than 10, so the serious unbalance numbers between histological types can’t have a good classification performance. We will continue to collect data and analyze this problem in our future study. We also add this in Discussion section. Comment 6: Finally, limitations of this study should be clearly stated. For instance, TNBC classification system integrating the expression profiles of mRNAs, but not IncRNAs, mutations or proteomic data. Response: We have added more discussion in Discussion section according this suggestion.

Round 2

Reviewer 1 Report

I'm satisfied with the revision.

Reviewer 2 Report

Authors have successfully replied all the concerns, thus I strongly suggest to accept the manuscript for publication in Genes in its actual form.